# Community-level cohesion without cooperation

**Mikhail Tikhonov**[1,2,3]*

[1]Center of Mathematical Sciences and Applications, Harvard University, Cambridge, United States; [2]Harvard John A Paulson School of Engineering and Applied Sciences, Harvard University, Cambridge, United States; [3]Kavli Institute for Bionano Science and Technology, Harvard University, Cambridge, United States

**Abstract** Recent work draws attention to community-community encounters ('coalescence') as likely an important factor shaping natural ecosystems. This work builds on MacArthur's classic model of competitive coexistence to investigate such community-level competition in a minimal theoretical setting. It is shown that the ability of a species to survive a coalescence event is best predicted by a community-level 'fitness' of its native community rather than the intrinsic performance of the species itself. The model presented here allows formalizing a macroscopic perspective whereby a community harboring organisms at varying abundances becomes equivalent to a single organism expressing genes at different levels. While most natural communities do not satisfy the strict criteria of multicellularity developed by multi-level selection theory, the effective cohesion described here is a generic consequence of resource partitioning, requires no cooperative interactions, and can be expected to be widespread in microbial ecosystems.

*For correspondence: tikhonov@fas.harvard.edu

**Competing interests:** The author declares that no competing interests exist.

## Introduction

Over the last decade, the sequencing-driven revolution in microbial ecology unveiled the staggering complexity of microbial communities that shape the health of our planet, and our own (*Caporaso et al., 2011*; *Lozupone et al., 2012*; *Human Microbiome Project Consortium, 2012*; *Gilbert et al., 2014*). These ecosystems routinely harbor hundreds of species of microorganisms, the vast majority of which remain poorly characterized. This makes the bottom-up approach to their modeling extremely challenging (*Greenblum et al., 2013*; *Bucci and Xavier, 2014*; *Ji and Nielsen, 2015*), prompting the question of whether some effective, top-down theory of the community as a whole might be a more viable alternative (*Doolittle and Zhaxybayeva, 2010*; *Borenstein, 2012*; *Greenblum et al., 2013*; *Bucci and Xavier, 2014*).

The need for a top-down approach is highlighted by multiple experimental observations. The microscopic species-level composition of independently assembled communities is highly variable even in similar environments; in contrast, the community metagenome (pathways carried by the population as a whole) appears to be more stable (*Human Microbiome Project Consortium, 2012*). Studies of obesity or inflammatory bowel disease indicate that these conditions are unlikely to be caused by specific 'pathogenic species' (*Major and Spiller, 2014*; *Mathur and Barlow, 2015*); similarly, the healthy human microbiota exhibits no core set of 'healthy' microorganisms (*Human Microbiome Project Consortium, 2012*). Thus, the 'healthy' and 'diseased' states of human-associated microbiota appear to be community-level phenotypic labels that may not always be traceable to specific community members.

Remarkably, the behavior of such macroscopically defined states can be productively studied even as the microscopic details remain unclear: thus, studies report on 'lean microbiota' outcompeting 'obese microbiota' in mice (*Ridaura et al., 2013*), or on the efficacy of fecal matter transplant in

**eLife digest** Microbes live in us and on us. They are tremendously important for our health, but remain difficult to understand, since a microbial community typically consists of hundreds of species that interact in complex ways that we cannot fully characterize. It is tempting to ask whether one might instead characterize such a community as a whole, treating it as a multicellular "super-organism". However, taking this view beyond a metaphor is controversial, because the formal criteria of multicellularity require pervasive levels of cooperation between organisms that do not occur in most natural communities.

In nature, entire communities of microbes routinely come into contact – for example, kissing can mix together the communities in each person's mouth. Can such events be usefully described as interactions between community-level "wholes", even when individual bacteria do not cooperate with each other? And can these questions be asked in a rigorous mathematical framework?

Mikhail Tikhonov has now developed a theoretical model that shows that communities of purely "selfish" members may effectively act together when competing with another community for resources. This model offers a new way to formalize the "super-organism" metaphor: although individual members compete against each other within a community, when seen from the outside the community interacts with its environment and with other communities much like a single organism.

This perspective blurs the distinction between two fundamental concepts: competition and genetic recombination. Competition combines two communities to produce a third where species are grouped in a new way, just as the genetic material of parents is recombined in their offspring.

Tikhonov's model is highly simplified, but this suggests that the "cohesion" seen when viewing an entire community is a general consequence of ecological interactions. In addition, the model considers only competitive interactions, but in real life, species depend on each other; for example, one organism's waste is another's food. A natural next step would be to incorporate such cooperative interactions into a similar model, as cooperation is likely to make community cohesion even stronger.

treating *Clostridium difficile* infections, whereby a 'healthy' community overtakes the 'diseased' state (*Bakken et al., 2011*). Both examples can be conceptualized as community-level competition events, termed 'community coalescence'. Although poorly understood, such events are widespread in natural microbial ecosystems and likely play a major role shaping their structure (*Rillig et al., 2015*). Intriguingly, Rillig *et al.* argue that coalescing communities often appear to be "interacting as internally integrated units rather than as a collection of species that suddenly interact with another collection of species" (*Rillig et al., 2015*).

Although comparing a community to a functionally integrated 'superorganism' is a recurring metaphor (*Shapiro, 1998*; *West et al., 2006*), a well-established body of theory cautions against using such terms loosely (*Gardner and Grafen, 2009*). The formal criteria under which a group of organisms can be considered a 'multicellular whole' have been extensively discussed in the context of multi-level selection theory (MLS) (*Okasha, 2008*). At the very least, the established notions of group-level individuality and 'organismality' crucially rely on cooperative traits of group members (*Buss, 1987*; *Michod, 1999*; *Michod and Nedelcu, 2003*). As a result, the formal applicability of the 'superorganism' perspective appears to be severely restricted, as pervasive cooperation between members must first be demonstrated. In particular, the microbiota inhabiting the human gut is extremely unlikely to satisfy such criteria.

However, the utility of a macroscopic community-level perspective, and its ability to predict the outcome of competition between communities, need not hinge on whether they constitute a valid level of selection in the strict sense of MLS. It is well known that performance of a species is dependent on community context (*Davis et al., 1998*; *McGill et al., 2006*; *McIntire and Fajardo, 2014*): for example, niche-packed communities (*MacArthur, 1969*; *Roughgarden, 1976*) are more resistant to invasion (*Levine and D'Antonio, 1999*). Building on these ideas, the present work extends the classical model of MacArthur (*MacArthur, 1969*) to construct a simple adaptive dynamics framework

that describes co-evolution in multi-species communities (*Roughgarden, 1976*; *Geritz et al., 1998*; *Nurmi and Parvinen, 2008*) and allows investigating the phenomenon of 'community coalescence' in a minimal theoretical setting. The central result is a mathematically precise analogy established between a community whose members can change in abundance and an individual organism whose pathways can modulate in expression. This analogy concerns the manner in which a community interacts with its environment and with other communities; it does not investigate reproduction, and so does not constitute multicellularity in the established sense of the term (*Okasha, 2008*). Rather than being a limitation, this expands the potential applicability of the top-down perspective advocated here. While the criteria of 'true multicellularity' are too stringent to apply to most natural communities, the phenomenon described in this work is a generic consequence of ecological interactions in a diverse ecosystem and requires no cooperative behavior or 'altruism' (*Gardner and Grafen, 2009*).

## Methods

### The metagenome partitioning model

To investigate community coalescence in the simplest theoretical setting, consider the following model for resource competition in a diverse community. It is closely related to MacArthur's model of competitive coexistence on multiple resources (*MacArthur, 1969*); see *Supplementary file 1*, section A.

Consider a community in a habitat where a single limiting resource exists in $N$ forms ('substrates' $i \in \{1 \ldots N\}$) denoted $A$, $B$, etc. For example, this could be carbon-limited growth in an environment with $N$ carbon sources, or a community limited by availability of electron acceptors in an environment with $N$ oxidants. The substrates can be utilized with 'pathways' $i$ (one specialized pathway per substrate). A species is defined by the pathways that it carries (similar, for example, to the approach of *Levin et al., 1990*). There is a total of $2^N - 1$ possible species in this model; they will be denoted using a binary vector indicating pathway presence/absence $\vec{\sigma} \equiv \{\sigma_i\} = \{1, 1, 0, 1, \ldots\}$, or by a string listing all substrates they can use, e.g. 'species $\underline{ABD}$' (the underline distinguishes specialist organisms such as $\underline{A}$ from the substrate they consume, in this case $A$). Let $n_{\vec{\sigma}}$ be the total abundance of species $\vec{\sigma}$ in the community, and let $T_i$ be the total number of individuals capable of utilizing substrate $i$ (*Figure 1*):

$$T_i \equiv \sum_{\vec{\sigma}} n_{\vec{\sigma}} \sigma_i.$$

Assume a well-mixed environment, so that each of these $T_i$ individuals gets an equal share $R_i/T_i$ of

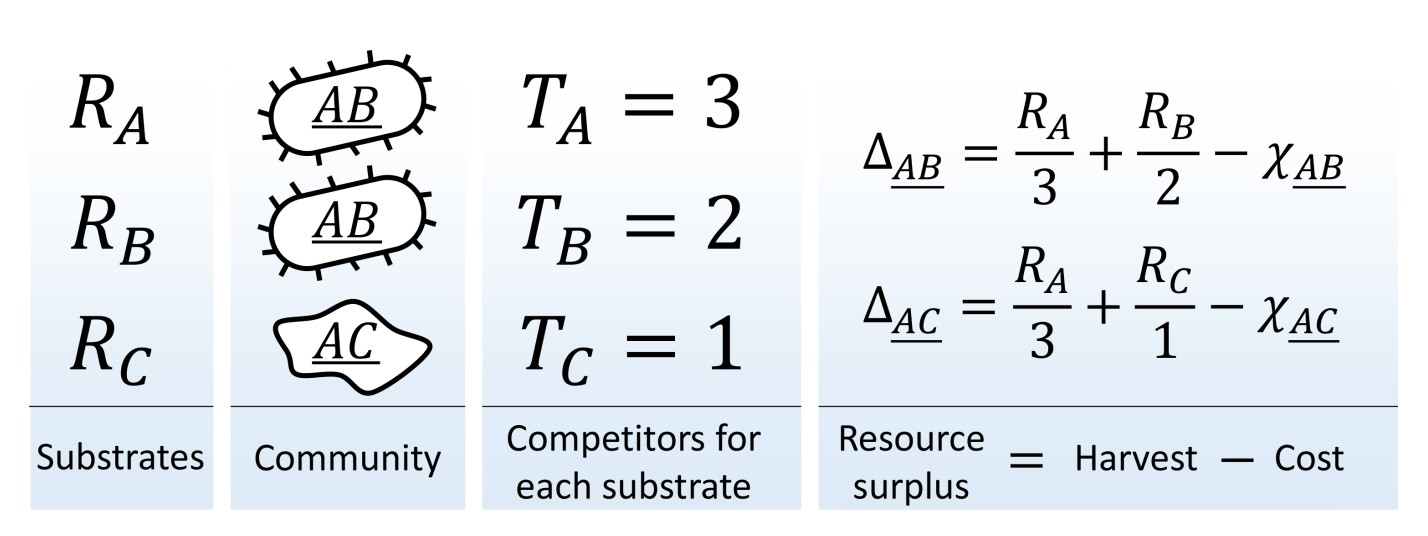

**Figure 1.** The metagenome partitioning model. Organisms are defined by the pathways they carry. The benefit from each substrate is equally partitioned among all organisms who can use it, and population growth/death of each species is determined by the resource surplus it experiences.

the total benefit $R_i$ (carbon content, oxidation power; etc.) available from substrate $i$ ('scramble competition'). Any one substrate is capable of sustaining growth, but accessing multiple cumulates the benefits. The population growth/death rate of species $\vec{\sigma}$ will be determined by the *resource surplus* $\Delta$ experienced by each of its individuals:

$$\Delta_{\vec{\sigma}} = \sum_i \sigma_i \frac{R_i}{T_i} - \chi_{\vec{\sigma}}. \tag{1}$$

Here, the first term is the benefit harvested by all carried pathways, and the second represents the maintenance costs of organism $\vec{\sigma}$. These costs summarize all the biochemistry that makes different species more or less efficient at processing their resources. For simplicity, let these costs be random:

$$\chi_{\vec{\sigma}} = \chi_0 |\vec{\sigma}| (1 + \epsilon \xi_{\vec{\sigma}}). \tag{2}$$

Here, $\chi_0$ is a constant (the average cost per pathway), $\xi_{\vec{\sigma}}$ is a random variable chosen once for each species and drawn out of the standard normal distribution (truncated to ensure $\chi_{\vec{\sigma}} > 0$), the 'cost scatter' $\epsilon$ sets the magnitude of cost fluctuations, and $|\vec{\sigma}| \equiv \sum_i \sigma_i$ is the number of pathways carried by the species: expressing more pathways incurs a higher cost (in this simple model, carrying and expressing a pathway is synonymous). The cost function (2) ensures that neither specialists nor generalists are systematically favored in competition (see below).

The resource surplus $\Delta_{\vec{\sigma}}$ is used to generate biomass. The simplest approach is to equate the biomass of an organism with its cost, so that the total biomass of a species is $\chi_{\vec{\sigma}} n_{\vec{\sigma}}$, and the dynamics of the model is given by:

$$\tau_0 \chi_{\vec{\sigma}} \frac{dn_{\vec{\sigma}}}{dt} = g_{\vec{\sigma}}(\{n_{\vec{\sigma}}\}) \equiv n_{\vec{\sigma}} \Delta_{\vec{\sigma}}. \tag{3}$$

The constants $\chi_0$ and $\tau_0$ set the units of resource and time. It is worth noting that a different choice for the biomass of each species would only change transient dynamics, but not the outcome of their competition: the equilibrium state where $\frac{dn_{\vec{\sigma}}}{dt} = 0$ (see **Supplementary file 1**, section B).

The approach taken here purposefully ignores multiple factors, most notably trophic interactions or any other form of organism inter-dependence. This is intentional: it ensures that the interaction matrix

$$M_{ab} \equiv \frac{\partial g_{\vec{\sigma}_a}}{\partial n_{\vec{\sigma}_b}}$$

has no positive terms, that is, the setting is purely competitive (indices $a$, $b$ label species, and $g_{\vec{\sigma}}$ is defined as the right-hand side of **Equation 3**). This helps underline that the whole-community behavior exhibited below is a generic consequence of resource partitioning, and requires no explicitly cooperative interactions.

Other simplifications include the assumption of deterministic dynamics and a well-mixed environment. Although stochasticity and spatial structure are tremendously important in most contexts, the simplified model adopted here provides a convenient starting point and makes the problem tractable analytically.

This work will investigate coalescence of communities that originate and remain in similar environments, for example, transfer of oral communities by kissing (**Kort et al., 2014**) as opposed to invasion of microbes from the mouth into the gut (**Qin et al., 2014**). Imagine a collection of islands (or patches) labeled by $\alpha$, each harboring a community $\mathcal{C}_\alpha$ experiencing the same environment. The next section investigates the within-island dynamics (3) to establish some key properties that make this simplified model particularly convenient for our purposes. Specifically, let $\Omega(\mathcal{C})$ denote the set of species present at non-zero abundance in a community $\mathcal{C}$. It will be shown that under the dynamics (3), any community $\mathcal{C}$ will eventually converge to a stable equilibrium, uniquely determined by the set $\Omega(\mathcal{C})$. At this equilibrium, certain species establish at a non-zero abundance, while others 'go extinct', exponentially decreasing towards zero. Importantly, the set of survivors will depend only on the identity of the initially present species, and not on their initial abundance. Thus a community $\mathcal{C}_1$ coalescing with $\mathcal{C}_2$ will yield the same community $\mathcal{C}_*$ irrespective of the initial mixing ratios. While obviously a simplification, this makes the metagenome partitioning model an especially convenient

starting point to build theoretical intuition about community-community interactions before more general situations can be studied, for example, numerically.

These properties are established in the next section; the following section then turns to the main subject of this work, namely coalescence events between islands.

## Single-island adaptive dynamics: intrinsic species performance and a community-level objective function

Consider $N = 10$ equiabundant substrates, and one random realization of organism costs with scatter $\epsilon = 10^{-3}$. (MATLAB scripts (MATLAB, Inc.) performing simulations and reproducing all figures are available as *Supplementary file 2*). The numerical simulation of competition between all 1023 possible species, initialized at equal abundance, results in the equilibrium state depicted in *Figure 2A*, *Supplementary file 2*. In this example, it consists of nine species. It is natural to ask: for a given initial set of competitors, what determines the species that survive?

In the present model, the only intrinsic performance characteristic of a species is its cost per pathway. Consider an assay whereby a single individual of species $\vec{\sigma}$ is placed in an environment with no other organisms present, and, for simplicity, an equal supply of all substrates $R_i = R$. The initial population growth rate in this chemostat is given by:

$$\left.\frac{dn_{\vec{\sigma}}}{dt}\right|_{t=0} = \frac{1}{\tau_0 \chi_{\vec{\sigma}}}\left[\sum_i R_i \sigma_i - \chi_{\vec{\sigma}}\right] = \frac{1}{\tau_0}\left[R\frac{|\vec{\sigma}|}{\chi_{\vec{\sigma}}} - 1\right],$$

and abundance eventually equilibrates at $n_{\vec{\sigma}} = R|\vec{\sigma}|/\chi_{\vec{\sigma}}$. Both these quantities characterize performance of species $\vec{\sigma}$ (the term 'fitness' is avoided as it is a micro-evolutionary concept that, strictly speaking, should be defined only within individuals of one species), and both are determined by the inverse cost per pathway. Define the 'individual' performance measure of species $\vec{\sigma}$ as

$$f_{\vec{\sigma}} \equiv \chi_0\frac{|\vec{\sigma}|}{\chi_{\vec{\sigma}}} - 1. \tag{4}$$

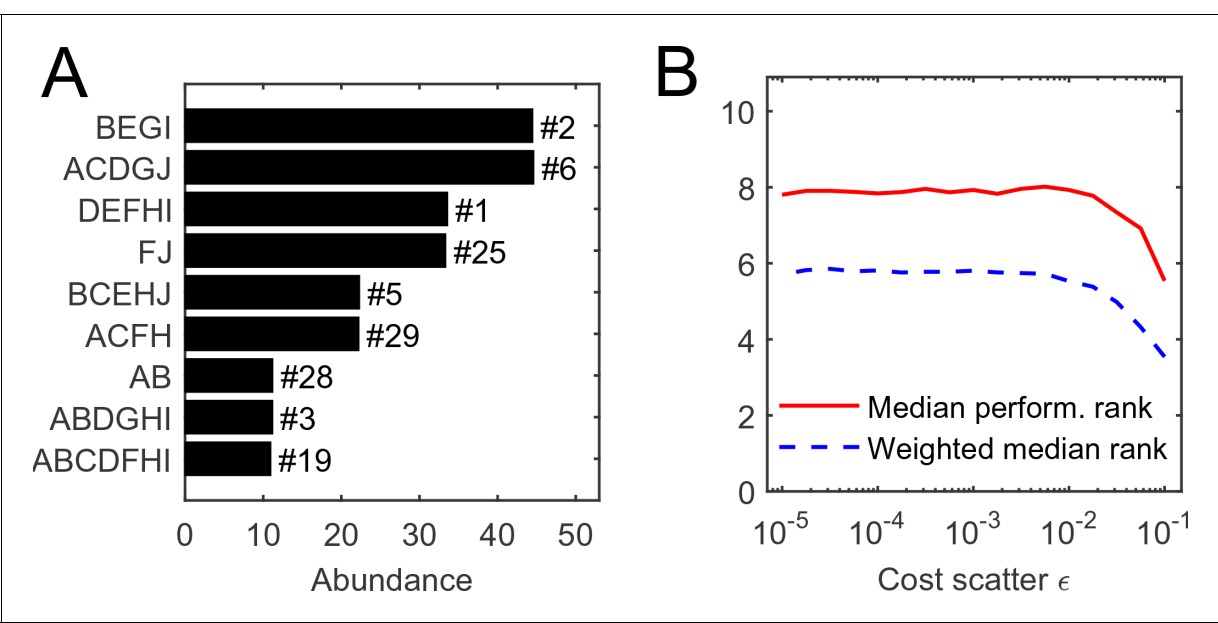

**Figure 2.** The individual performance rank of a species (its cost per pathway) is predictive of its survival and abundance in a community. (**A**) Community equilibrium for $N = 10$ substrates with abundance $R_i/\chi_0 = 100$ and one particular random realization of organism costs with scatter $\epsilon = 10^{-3}$. Species are ordered by abundance and labeled by the pathways they carry. Also indicated is the individual performance rank; all surviving species were within the top 30 (out of 1023). (**B**) The median individual performance rank of survivors, weighted (dashed) or not weighted (solid) by abundance. Curves show mean over 100 random communities for each value of cost scatter $\epsilon$; the standard deviation across 100 instances is stable at approximately 40% of the mean for both curves, independently of $\epsilon$ (not shown to reduce clutter).

This definition is convenient as it makes $f_{\vec{\sigma}}$ a dimensionless quantity of order $\epsilon$. Under the cost model (2), the performance ranking of species is random, set by the random realization of the costs $\xi$: $f_{\vec{\sigma}} = \frac{1}{1+\epsilon\xi_{\vec{\sigma}}} - 1 \approx -\epsilon\xi_{\vec{\sigma}}$. This model was chosen so that no group of species has an obvious advantage. A different cost function would effectively reduce the pool of competitors, excluding certain (prohibitively expensive) species from the start.

Predictably, this performance ranking is correlated with the success of a species in a community, but not very well (*Figure 2*). The equilibrium depicted in panel A predominantly consists of top-ranked species, and the median performance rank of surviving species is consistently low across a range of values of the cost scatter $\epsilon$ (panel B). This median rank becomes even lower if the median is weighted by a species' abundance at equilibrium, indicating that top-ranked species tend to be present at higher abundance (*Davis et al., 1998*; *Birch, 1953*). Still, at the equilibrium shown in *Figure 2A*, the species ranked fourth in intrinsic performance went extinct, but six others ranked as low as #29 remained present.

These observations reflect the well-known fact that the success of a species is context-dependent and observing a species in isolation does not measure its performance in the relevant environment (*McGill et al., 2006*; *McIntire and Fajardo, 2014*). In the model described here, the context experienced by all species is fully encoded in the vector of 'harvests', that is, the benefit an organism receives from carrying pathway $i$:

$$H_i \equiv R_i/T_i. \tag{5}$$

A growing demand for substrate $i$ (increasing $T_i$) depletes its availability, in the sense that $H_i$ is reduced. Consider the three-substrate world depicted in *Figure 1*, and assume that $\underline{AB}$ is the highest-performing species with a very low cost. As $\underline{AB}$ multiplies, it depletes resources $A$ and $B$. As a result, the final equilibrium is highly likely to include the specialist organism $\underline{C}$, even if its cost is relatively high, and under other circumstances (if $\underline{AB}$ were less fit) it would have yielded to $\underline{AC}$ or $\underline{BC}$.

Conveniently, in the model described here, these complex effects studied by niche construction theory can be summarized in a single community-level objective function. The dynamics (3) possess a Lyapunov function, i.e. a quantity that is increasing on any trajectory of the system (compare to *MacArthur, 1969*):

$$F = \frac{1}{R_{\text{tot}}}\left(\sum_i R_i \ln\frac{T_i}{R_i/\chi_0} - \sum_{\vec{\sigma}}\chi_{\vec{\sigma}}n_{\vec{\sigma}} + R_{\text{tot}}\right). \tag{6}$$

Here, $R_{\text{tot}}$ is a constant introduced for later convenience. Specifically, set $R_{\text{tot}} = \sum_i R_i$; this choice ensures that close to community equilibrium, $F$ is also of order $\epsilon$ (see *Supplementary file 1*, section C). This function has the property that $R_{\text{tot}}\frac{\partial F}{\partial n_{\vec{\sigma}}} = \Delta_{\vec{\sigma}}$ (the resource surplus), and therefore

$$\frac{dF}{dt} = \sum_{\vec{\sigma}}\frac{\partial F}{\partial n_{\vec{\sigma}}}\frac{dn_{\vec{\sigma}}}{dt} = \sum_{\vec{\sigma}}\frac{n_{\vec{\sigma}}(\Delta_{\vec{\sigma}})^2}{R_{\text{tot}}\chi_0\tau_0} > 0$$

Thus, $F$ is indeed monotonically increasing as the system is converging to equilibrium. To illustrate this, *Figure 3A* shows 10 trajectories of ecological dynamics for the same system as in *Figure 2A*, starting from random initial conditions (with all species present; see *Supplementary file 1*, section H). Far from equilibrium, while most high-cost species are being eliminated by competitors, the mean intrinsic performance of surviving organisms and $F$ increase together (*Figure 3A*, inset), confirming that intrinsic performance is a useful predictor. However, as equilibrium is approached, community-induced changes in substrate availability $H_i$ reduce the relevance of the original performance ranking, which was measured in the 'wrong' environment; previously successful species can be driven to extinction (*Figure 3B*). The set $\{H_i\}$ at equilibrium characterizes the environment the surviving species had 'carved' for themselves. The performance rank ordering will be all the more sensitive to the environment $\{H_i\}$, the smaller the scatter of intrinsic organism costs $\epsilon$. Therefore, the role of this parameter is to tune the relative magnitude of intrinsic and environment-dependent factors in determining a species' fate. So far, $\epsilon$ was fixed at $10^{-3} \approx 2^{-N}$, and *Figure 2B* shows that for small cost scatter $\epsilon$, the structure of the final equilibria does not significantly depend on this parameter (see *Supplementary file 1*, section D). The large-$\epsilon$ regime will be discussed later.

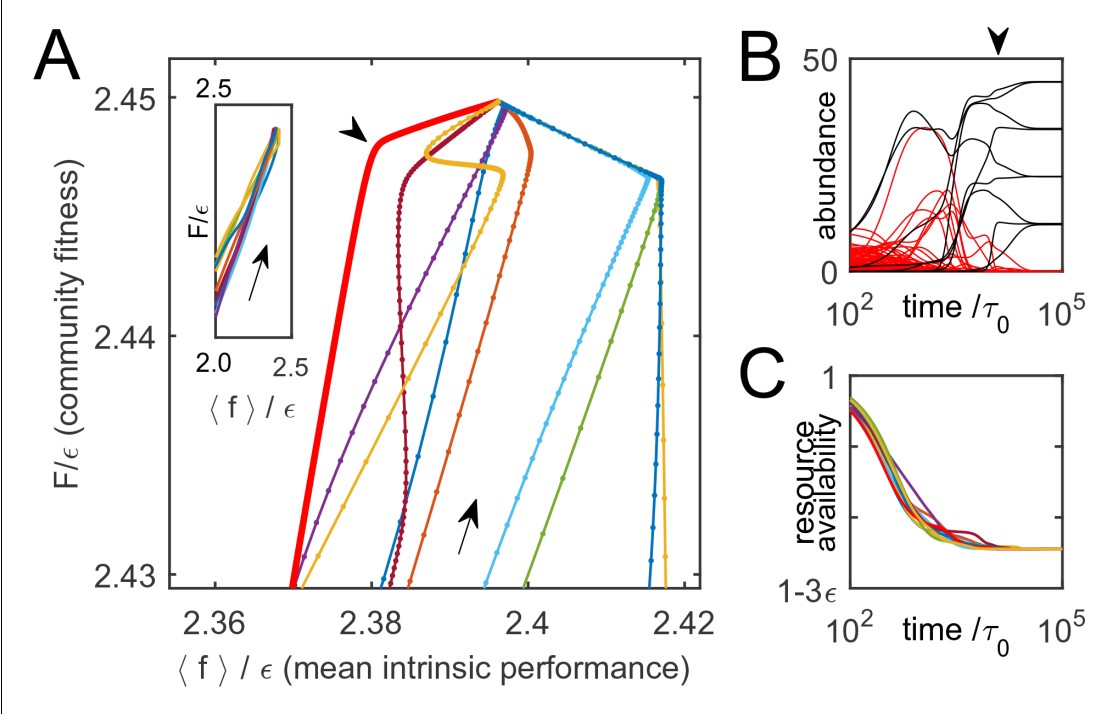

**Figure 3.** Community dynamics maximize a global objective function. (**A**) 10 trajectories of an example system, starting from random initial conditions and converging to the equilibrium depicted in *Figure 2A*. Direction of dynamics indicated by arrows. Far from equilibrium, mean intrinsic performance of members (weighted by abundance) and the community-level function $F$ increase together (inset; data aspect ratio as in the main panel). Close to equilibrium, intrinsic performance loses relevance. (**B**) Time traces of species' abundance for one community trajectory (thick red line in **A**). Arrowheads in panels A and B indicate matching time points. Species that eventually go extinct shown in red; many enjoy transient success. (**C**) The complex dynamics of panel B is driven by the simple objective to efficiently deplete all substrates simultaneously, encoded in $F$. Shown is mean availability of the 10 substrates, for each trajectory of panel A.

Each of the trajectories in *Figure 3A* converges to the same equilibrium (depicted in *Figure 2A*). This is because $F$ is convex and bounded from above (see *Supplementary file 1*, section A). Therefore, for every set of species $\Omega$, any community restricted to these species will always reach the same (stable) equilibrium, corresponding to the unique maximum of $F$ within the subspace where only species of $\Omega$ are allowed non-zero abundance. This maximum will often be at the border of this subspace, corresponding to the extinction of some species.

Under the dynamics (3), no new species can 'appear' if their original abundance was zero. Imagine, however, that on each island, a rare mutation (or migration) occasionally introduces a random new species; if it can invade, the community transitions to a new equilibrium and awaits a new mutation. This process of adaptive dynamics defines the evolution of each island, and can be seen as a mesoscopic population genetics model for a multi-species community evolving through horizontal gene transfer (loss/acquisition of whole pathways). For each island, $F$ is monotonically increasing throughout its evolution. Indeed, $F$ is continuous and non-singular in all $n_{\vec{\sigma}}$, so introducing an invader at a vanishingly small abundance will leave $F$ unchanged, and the subsequent convergence to a new equilibrium is a valid trajectory of ecological dynamics on which $F$ increases.

What is the intuitive meaning of the function $F$ that is being optimized by the community? It is easy to show that $\sum_{\vec{\sigma}} \chi_{\vec{\sigma}} n_{\vec{\sigma}} = \sum_i R_i$ at any equilibrium (total demand matches the total supply; see *Supplementary file 1*, section C). Therefore, for a community at equilibrium, $F$ characterizes its ability to deplete substrates:

$$F = -\frac{\sum_i R_i \ln H_i}{R_{\text{tot}}} + \text{const} \tag{7}$$

If all substrates are equiabundant for simplicity, maximizing $F$ is equivalent to minimizing $\sum_i \ln H_i$.

The optimization principle that appears in this model is therefore a generalization of Tilman's $R^*$ rule (*Tilman, 1982*). In the classic form, this rule states that for a single limiting resource, the unique winning species is the one capable of depleting this resource to the lowest concentration. However, if the limiting resource can be harvested from multiple substrates, as considered here, multiple species may coexist; the winning community is the one that is most efficient at depleting all substrates simultaneously, weighted as described in *Equation (7)*. This is illustrated in *Figure 3C*. While the time trajectories of individual species may be highly complex (*Figure 3B*), the net effect of these dynamics is to deplete substrate availability down to the lowest concentrations capable of sustaining a population (see also Figure S1 in *Supplementary file 1*, section H).

The following sections will argue that $F$ can be thought of as community-level 'fitness', but this term will not be used until justification is provided.

## The community-level function $F$ predicts the outcome of community coalescence

Consider now a coalescence event whereby the equilibrium communities from two islands $\mathcal{C}_\alpha$ and $\mathcal{C}_\beta$ are brought into contact; as established above, the resulting community $\mathcal{C}_*$ will not depend on the details of the mixing protocol. If none of the species from island $\beta$ could invade the community $\mathcal{C}_\alpha$, then $\mathcal{C}_* = \mathcal{C}_\alpha$ and the community $\mathcal{C}_\alpha$ is the clear winner. In general, however, the space of competition outcomes is richer than merely one community taking over: both competitors $\mathcal{C}_\alpha$, $\mathcal{C}_\beta$ can contribute to $\mathcal{C}_*$, but can be more or less successful at doing so, contributing more or fewer species. What makes a community more likely to be successful?

The community on each island $\alpha$ constructs its own environment, establishing certain levels of substrate availability $\{H_i^{(\alpha)}\}$. When species from island $\alpha$ are introduced onto island $\beta$, they are exposed to a random new environment, and the equilibrium environment $\{H_i^*\}$ that the coalescence survivors will have created for themselves will be different still. Although success of a species is environment-dependent, for a random environment, $f_{\vec{\sigma}}$ as defined above remains the best available performance predictor. One may therefore expect that the more successful community should be the one with more high-performance species. On the other hand, we also found that the ultimate equilibrium community that cannot be invaded by *any* species does not consist of species with the highest intrinsic performance, but corresponds to the global maximum of $F$. This suggests that the community-level function $F$ should be the better predictor of the competition outcome. If so, it could be said to characterize the 'collective fitness' of a community (in the restricted, purely competitive, rather than reproductive, sense).

To settle the competition between these two hypotheses, the following procedure was implemented. For $N = 10$ substrates, and a given random realization of the cost structure $\xi$ with scatter $\epsilon = 10^{-3}$, $M = 50$ random species were selected to allow for an exhaustive sampling of sub-communities (the results reported below do not significantly depend on this choice). This set was used to construct all $\binom{50}{4} = 230300$ possible combinations of $k = 4$ species, 104,006 of which constituted fully functional communities with all $N = 10$ pathways present; these were independently equilibrated. The putative collective fitness $F$ of these communities, and the mean individual performance of their members, is shown in *Figure 4A*. This procedure puts at our disposal multiple examples of communities where the two performance measures are both high, both low, or one is high while the other is low (the quadrants highlighted in *Figure 4A*). Competing pairs of communities drawn from these pools will make it possible to determine which of the two factors, individual performance of a species $f_{\vec{\sigma}}$ or the collective fitness $F$ of its native community, can better predict its post-coalescence survival.

To begin, consider the competition between the cyan and magenta quadrants (I and III, respectively). Communities from the magenta quadrant are predicted to be more fit, both in the collective sense and as measured by the average intrinsic performance of members. Therefore, one expects that the magenta (III) communities should, on average, be more successful in pairwise competitions. To confirm this, *Figure 4B* presents the results of an 'elimination assay' competing communities from these quadrants. Five hundred random pairs were drawn, and correspond to columns in *Figure 4B*. For each pair, species from both communities (up to 8 each time) were equilibrated together; the rows in *Figure 4B* correspond to these species, ordered by individual performance rank: high (top) to low (bottom). For each species that went extinct during equilibration, its

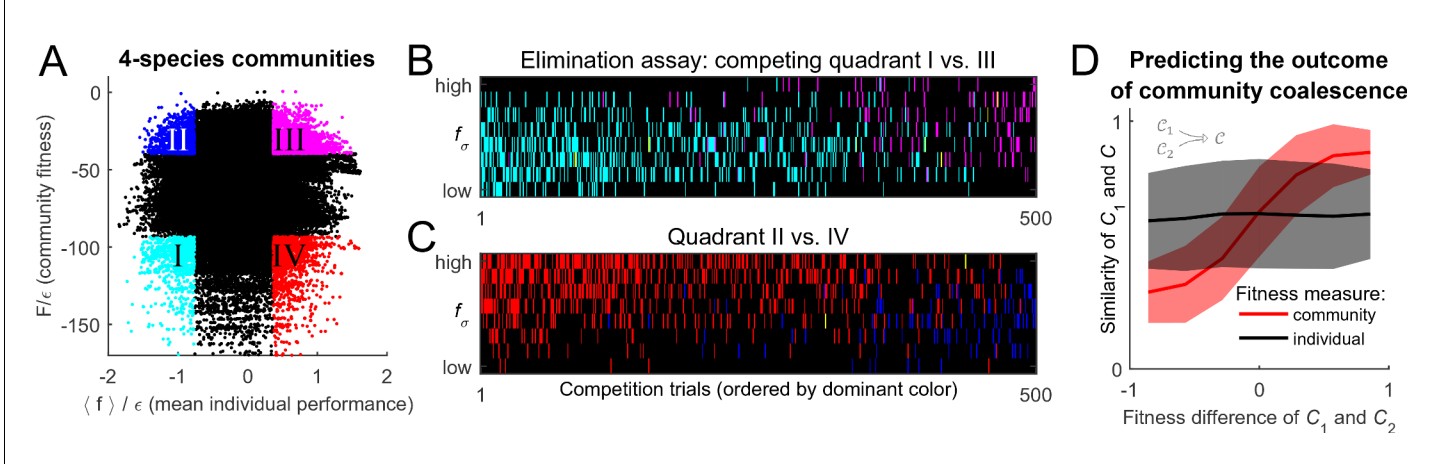

**Figure 4.** Community fitness is more predictive of competition outcome than the intrinsic performance of its members. (A) Community fitness $F$ vs mean intrinsic performance $\langle f_{\vec{\sigma}} \rangle$ of its members, measured in units of cost scatter $\epsilon$, for 104,006 communities with four species (see text). Communities in which both characteristics are in the top or bottom 10% are highlighted. (B) Elimination assay competing quadrants I (cyan) vs III (magenta). Five hundred randomly drawn community pairs (columns) were jointly equilibrated, with up to eight species each time (rows; ordered by $f_{\vec{\sigma}}$). For each species that went extinct during equilibration, the corresponding cell in the table is colored by the species' provenance. As expected, most eliminated species were from the less fit cyan communities (there are more cyan cells than magenta). These species also had lower $f_{\vec{\sigma}}$ (most colored cells are in the lower half of the table). (C) Same, competing quadrants II (blue) vs IV (red). The dominant color is now red: most eliminated species were from red communities, and went extinct despite having higher $f_{\vec{\sigma}}$ (most colored cells are in the upper half of the table). Columns ordered by dominant color. (D) Community similarity $S(C_1, C)$ for a coalescence event depicted in the cartoon (inset), computed for $10^4$ random community pairs, as a function of fitness difference between competing communities. Fitness difference scaled to the maximum of 1 so both fitness measures can be shown in same axes. Shown is binned mean (7 bins) over communities with similar fitness difference (solid line) $\pm 1$ standard deviation (shaded).

provenance was identified ("did it come from the magenta or the cyan community?"), and the corresponding rectangle in *Figure 4B* was colored accordingly; in the rare cases when the eliminated species was originally present in both communities, it was colored yellow. The dominant color in *Figure 4B* is cyan, confirming that the cyan communities are typically less successful at contributing their members to the final equilibrium. Note also that the colored entries are predominantly located in the bottom half of the table: the eliminated species tend to also have lower intrinsic performance than their more successful competitors. This is the expected result.

Now, consider the competition between blue and red quadrants (II and IV). An elimination assay conducted in an identical manner is presented in *Figure 4C*. Now the colored entries are predominantly red and occupy the *top* half of the table. In other words, members of the red communities are being outcompeted despite the fact that their intrinsic performance is higher: the individual performance of a species is less predictive of its ability to survive coalescence than the collective fitness of the community of which it was part.

Finally, $10^4$ random community pairs from the pool of *Figure 4A* (not restricted to any quadrant) were competed. Define community similarity for $C_1 \equiv \{n_{1\vec{\sigma}}\}$ and $C_2 \equiv \{n_{2\vec{\sigma}}\}$ as the normalized scalar product of their species abundance vectors:

$$S(C_1, C_2) = \frac{\sum_{\vec{\sigma}} n_{1\vec{\sigma}} n_{2\vec{\sigma}}}{\sqrt{\sum_{\vec{\sigma}} n_{1\vec{\sigma}}^2} \sqrt{\sum_{\vec{\sigma}} n_{2\vec{\sigma}}^2}}.$$

For each of the $10^4$ coalescence instances $C_1 + C_2 \mapsto C_*$, *Figure 4D* plots the similarity $S_1 \equiv S(C_1, C_*)$ as a function of fitness difference between 'parent' communities $C_1$ and $C_2$. It comes as no surprise (*cf. Figure 2*) that the predictive power of the mean individual performance is extremely weak (black line). In contrast, community fitness is a strong predictor: the larger the difference in community fitness, the stronger the similarity between the post-coalescence community and its more fit parent (red line). In the mathematical framework developed here, the observation that coalescing communities appear to be 'interacting as coherent wholes' acquires a precise formulation. Without implying

the emergence of any new level of selection, and without invoking any cooperative traits, we observe that community coalescence can be usefully described as an interaction between two entities, characterized macroscopically at the whole-community level.

## The 'community as an individual' metaphor becomes exact

Consider now an external observer who is denied direct microscopic access to community composition, and is able to perform only 'metagenomic' (or, rather, 'metaproteomic') experiments, measuring the community-wide pathway expression $\vec{T} = \{T_i\}$ in response to substrate influx $\vec{R} = \{R_i\}$.

First, consider an island $\alpha_G$ harboring a single species: the complete generalist $\vec{\sigma}_G = \{1, 1 \ldots 1\}$. Its abundance at equilibrium will be $n_G = T_i = R_{\text{tot}}/\chi_G$. Although substrates may be supplied in varying abundance, the island $\alpha_G$ can only express all pathways at the same level.

Another island $\alpha_S$ might harbor a community of perfect specialists: $\underline{A} = \{1, 0, 0 \ldots\}$, $\underline{B} = \{0, 1, 0 \ldots\}$, etc. Faced with an uneven supply of substrates, this island will adjust expression levels $T_i$ to precisely track the supply vector $R_i$, so that $T_i = R_i/\chi_i$, where $\chi_i$ is the cost of the respective specialist. For an external observer whose toolkit is limited to investigating the mapping $\vec{R} \mapsto \vec{T}$, the specialists' island $\alpha_S$ is formally indistinguishable from an organism who can sense its environment and up-regulate or down-regulate individual pathways.

Such perfect regulation is, however, costly: typically, $\underline{A}$, $\underline{B}$, etc. will not be the most cost-efficient combinations. As a result, allowing the community to evolve while holding $\vec{R}$ fixed, one will obtain a different multi-organism community $\mathcal{C}$. Unlike $\alpha_S$, it will generally be unable to respond to all environmental perturbations: for example, the nine-species equilibrium community of *Figure 2A* will necessariy be insensitive to some direction in the 10-dimensional space of substrate concentrations. Our external observer will conclude that evolution in a stable environment has traded some of the sensing capacity for the ability to fit a particular substrate influx with more efficient pathway combinations.

The model presented here can therefore be reinterpreted as a model for adaptive evolution of a single organism striving to better adjust its response $\vec{T}$ to the environment $\vec{R}$ it experiences. The model specifies how the genotype (patterns of pathway co-regulation) determines phenotype (the mapping $\vec{R} \mapsto \vec{T}$), and the competitive fitness $F$ is an explicit function of both the genotype and the environment (*Ribeck and Lenski, 2015*). To conclude this section, let us compute the community fitness $F$ of the single-species generalist community $\alpha_G$ for the case $R_i \equiv R$. Applying the definition (6), and using $T_i = n_G = NR/\chi_G$ one finds:

$$F = \frac{1}{\sum_i R_i} \left( \sum_i R_i \ln \frac{T_i}{R_i/\chi_0} - n_G \chi_G \right) + 1 = \ln \frac{N\chi_0}{\chi_G} = \ln(1 + f_G) \approx f_G$$

where $f_G$ is the individual performance (4) of organism $\sigma_G$, and the approximate equality holds because $f_G$ is of order $\epsilon$, assumed small. In other words, for a single-species community, the community fitness coincides with the individual performance of that species, reinforcing the emergent parallel between a community and an individual that had evolved an internal division of labor. This interpretation is specific to the particular model explored here, but within this model, the metaphor is mathematically exact.

## Community cohesion as a generic consequence of ecological interactions

It is important to contrast the results of the previous section with the notion of 'fitness decoupling' in multi-level selection theory (MLS). In MLS, a higher level of organization is recognized when a group of cooperating organisms acquires interests that are distinct from the self-interest of its members (*Okasha, 2008*). Here, competition always remains entirely 'selfish'. In each instance of community competition assayed in *Figure 4*, whenever some species invaded a community, it was because its fitness in *that particular environment* was higher than the fitness of species already present. In contrast to fitness decoupling, which requires special circumstances to evolve, the community-level cohesion described in this work is a generic consequence of the fact that organisms modify their environment, and that fitness is context-dependent (*Hay et al., 2004*; *McGill et al., 2006*; *McIntire and Fajardo, 2014*; *Ribeck and Lenski, 2015*).

The definition (4) corresponds to how we might experimentally measure fitness, by placing an organism in a 'typical' environment it is believed to experience. In the model described here, this typical environment is often an excellent approximation: for a community at equilibrium with equi-abundant substrates $R_i = R$, the total community-wide expression of each pathway is roughly $T \approx R/\chi_0$, the same for all $i$. Nevertheless, even small deviations may be sufficient to induce substantial reordering of the relative performance rank of different species, in which case the context-dependent component of fitness can become dominant.

If this interpretation of the results of *Figure 4* is correct, then reducing the degree to which environmental perturbations affect relative fitness of individuals should lead to a tighter link between community fitness and individual species' performance. This prediction can be tested by increasing $\epsilon$, the parameter that determines the width of the distribution of organism costs. For example, consider a community where the substrate $A$ is disputed by only two organisms: $\underline{A}$ and $\underline{AB}$. Assume that $f_{\underline{A}} > f_{\underline{AB}}$, so that when substrates $A$ and $B$ are equally abundant, the species $\underline{A}$ displaces $\underline{AB}$(the resource $B$ is then consumed by some other species). Reducing the availability of substrate $A$ can reverse this outcome (if $A$ is absent, $\underline{AB}$ can still survive, but not $\underline{A}$). However, the larger the difference in intrinsic performance $f_{\underline{A}}$ and $f_{\underline{AB}}$, the more extreme such resource depletion would have to be. Therefore, increasing the intrinsic cost scatter $\epsilon$ will reduce the relative effect that changing environment has on fitness rank ordering. *Figure 5* repeats the analysis of *Figure 4A* for $\epsilon = 0.1$ (rather than $\epsilon = 10^{-3}$ used previously). As predicted, the collective fitness is now strongly associated with the performance of individuals. In fact, this is already apparent in *Figure 2B*: as $\epsilon$ is increased, the median fitness rank of survivors at the final equilibrium begins to reduce. At high $\epsilon$, it is increasingly true that high collective fitness is merely a reflection of high intrinsic performance of community members. Thus, *Figure 2B* documents a transition between a largely individualistic regime (at large

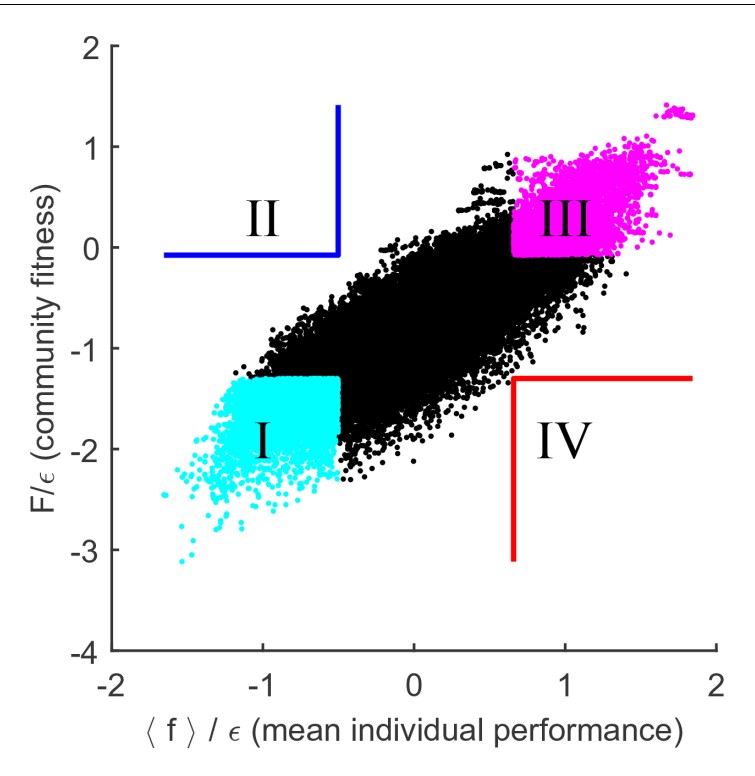

**Figure 5.** Cost scatter $\epsilon$ tunes the magnitude of community cohesion. Same as *Figure 4A*, for larger $\epsilon = 0.1$. Increasing the scatter of intrinsic costs $\epsilon$ reduces the relative importance of environment in determining the performance ranking of species. As a result, collective fitness of a community and the mean individual performance of its members remain strongly coupled. Defining quadrants as in *Figure 4A* leaves the blue and red quadrants empty.

$\epsilon$) and a regime where the genetically inhomogenous assembly of species increasingly acts 'as a whole', in the precise sense discussed in the previous section.

## Discussion

This work presented a theoretical framework where the analogy between a community harboring organisms at varying abundances, and an organism expressing genes at different levels, becomes an exact mathematical statement. A striking feature of this perspective is the blurred boundary between the notions of competition and genetic recombination (*Shapiro et al., 2012*; *Rosen et al., 2015*). Consider competition between organisms as an operation that takes two organisms and yields one:

$$\text{Competition}: \quad (\mathcal{O}_1, \mathcal{O}_2) \mapsto \mathcal{O}_*.$$

Traditionally, the space of outcomes is binary: one competitor lives, one dies, and the propensity to survive competition is called fitness. When competition between communities of organisms is considered, this definition must inevitably be generalized to allow $\mathcal{O}_*$ to be distinct from either of the original competitors. Such 'competitors', however, might be more aptly named 'parents'. In sexual reproduction, recombination allows a subset of the genes inherited from both parents to form progeny with potentially higher fitness; here, the competition between parent communities $\mathcal{C}_\alpha$ and $\mathcal{C}_\beta$ allows a subset of their members to regroup into a daughter community $\mathcal{C}_*$ with a higher collective fitness $F$. The parallel becomes especially clear if one imagines propagules of $\mathcal{C}_\alpha$ and $\mathcal{C}_\beta$ co-colonizing a fresh environmental patch.

Such member regrouping can be much more flexible than the rules of sexual recombination, but reduces to the latter in the particular case of communities with clearly demarcated functional guilds (e.g., consider competition between two communities that each has one plant, one pollinator, one herbivore, one carnivore, etc.). Long before the evolution of sex, such recombination would have allowed communities with divided labor to fix evolutionary novelty more efficiently than a clonal population of generalists. Although the metaphor of a genome as an 'ecosystem of genes' is not new (*Avise, 2001*), the framework presented here allows it to be formalized and investigated quantitatively (see also *Akin, 1979* and *Supplementary file 1* Section J).

The results in this work were derived within the simplified framework of a particular model where microscopic dynamics conveniently took the form of optimizing a community-level objective function. In general, of course, collective dynamics are almost never reducible to solving an optimization problem (*Akin, 1979*; *Hofbauer and Sigmund, 1998*; *Metz et al., 2008*). However, the effective cohesion of coalescing communities described here is merely a result of environment-dependent species' performance combined with the community shaping its own environment, a niche construction effect (*Scott-Phillips et al., 2014*) not specific to one modeling framework. A certain parallel can also be seen with the hypothesis that niche-packed communities may be more resistant to invasion (*MacArthur, 1955*; *Levine and D'Antonio, 1999*). In the model at hand, the existence of a global objective function made this phenomenon particularly easy to investigate; in a more general model, it would not be possible to quantify this effect with a single number (a 'community fitness'). Nevertheless, the qualitative result may be expected to persist, so that members of a co-evolved community with a history of coalescence would tend to have higher persistence upon interaction with a 'naive' community that had never been exposed to such events, as proposed by *Rillig et al. (2015)*. More work is required to verify the generality of this hypothesis.

The results presented here, derived in a purely competitive model, demonstrate that functional cohesion is conceptually separate from the discussions of 'altruism' and cooperation (*Gardner and Grafen, 2009*), except to the extent described by the formula 'enemy of my enemy is my friend' (indirect facilitation [*Levine, 1999*]). The latter can be seen as a form of cooperation (*Hay et al., 2004*), but is a generic phenomenon and is not vulnerable to 'cheaters'.

While the criteria of 'true multicellularity' are too stringent to apply to most natural communities, the phenomenon described in this work is a generic consequence of ecological interactions in a diverse ecosystem. Many effects omitted here can be expected to further contribute to such cohesion, especially co-evolved interdependence of organisms. If whole-community coalescence events are indeed a significant factor shaping the evolutionary history of microbial consortia, then

community-level cohesion of the type described here can be expected to be broadly relevant for natural ecosystems (*Doolittle and Zhaxybayeva, 2010*).

## Acknowledgements

I thank Ariel Amir, Michael P. Brenner, Andy Gardner, Jeff Gore, Miriam H. Huntley, Simon A. Levin, Anne Pringle, Ned S. Wingreen and David Zwicker for helpful discussions, and anonymous referees for their comments on the early version of the manuscript. I have no competing interests. This work was supported by the Harvard Center of Mathematical Sciences and Applications, and the Simons Foundation.

## Additional information

### Funding

| Funder | Author |
| --- | --- |
| Simons Foundation | Mikhail Tikhonov |
| Harvard Center of Mathematical Sciences and Applications | Mikhail Tikhonov |

The funders had no role in study design, data collection and interpretation, or the decision to submit the work for publication.

### Author contributions

MT, Conception and design, Acquisition of data, Analysis and interpretation of data, Drafting or revising the article

### Author ORCIDs

Mikhail Tikhonov, http://orcid.org/0000-0002-9558-1121

## Additional files

### Supplementary files

• Supplementary file 1. Technical details, numerical procedures, and relation to the model of MacArthur.

• Supplementary file 2. MatLab scripts reproducing all figures (optional data files with pre-computed simulation results included for faster figure plotting).

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
