## [Decision Letter]

Thank you for submitting your article "Community-level cohesion without cooperation" for consideration by *eLife*. Your article has been reviewed by three peer reviewers, including Jeff Gore and Carl Bergstrom, who is a member of our Board of Reviewing Editors, and the evaluation has been overseen by Ian Baldwin as the Senior Editor.

The reviewers have discussed the reviews with one another and the Reviewing Editor has drafted this decision to help you prepare a revised submission.

Summary:

The author has developed a simple and powerful model of ecological community formation. The reviewers appreciate the novelty and creativity of the approach, and value the model for its simplicity, its tractability, and its role as a potential null model for thinking about community formation. With appropriate revisions this paper merit publication in *eLife* and serve as a valuable contribution to the research community more broadly.

Essential revisions:

1) Please attempt to provide a better intuitive understanding of what contributes to community level fitness and how it "works" with respect to resource partitioning, redundancy, etc. Is there a simple description of what is the quantity maximized by the community?

For example, it would be useful to find ways to visualize how community competitive ability (*F*) relates to the distribution of resource processing among community members (e.g. a plot of some metric of resource overlap against *F*). For the section 3 experiments, any theory that could developed in this arena would be very welcome. Even a nice example would help. How does AB, CD, EFG, HIJ do against invaders, for example? I might intuit that it would be hard to beat what for its efficient resource partitioning, but do "cross linkages" e.g. ABCF, CDEI, EFG, GHIJA do even better because they help adjust to resource availability as discussed in section IV? Do these cross-linkages help because they trade off growth against stability to environmental perturbations, as discussed in section V?

2) Please at least discuss the sensitivity of the model results to the form of the cost function. For example, how would we expect adding an exponent to norm (σ) in the cost function (2) to change things? This would modulate something like the efficiency of adding additional pathways. Are the conclusions of this work dependent on this assumption that the cost is proportional to the biomass?

3) Throughout, terms could be better defined and explained in the text. In general, the paper is quite heavy in notation, some of which may even be unnecessary. Remind the reader periodically what the variables stand for. For example, each variable could have a name in English and that name could be used together with the symbol in the text. Though it's clear from context, I don't think the paper ever explains in words how *σ_i_*is to be interpreted. Nor does *g_σ_* seem to be defined. *H_i_^(α)^* is not clearly explained; it's an environment, but explain how parametrized. Similarly, the figure and figure legends could be made more self-explanatory. For example, in Figure 1 I had to go back to the main text to determine what *ε* was. This should be explained in words. In the interest of reaching the broadest possible set of readers perhaps consider providing a bit more explanation for technicalities such as what a Lyapunov function is.

---

## [Author Response]

*Essential revisions:*

*1) Please attempt to provide a better intuitive understanding of what contributes to community level fitness and how it "works" with respect to resource partitioning, redundancy, etc. Is there a simple description of what is the quantity maximized by the community?*

*For example, it would be useful to find ways to visualize how community competitive ability (F) relates to the distribution of resource processing among community members (e.g. a plot of some metric of resource overlap against F). For the section 3 experiments, any theory that could developed in this arena would be very welcome. Even a nice example would help. How does AB, CD, EFG, HIJ do against invaders, for example? I might intuit that it would be hard to beat what for its efficient resource partitioning, but do "cross linkages" e.g. ABCF, CDEI, EFG, GHIJA do even better because they help adjust to resource availability as discussed in section IV? Do these cross-linkages help because they trade off growth against stability to environmental perturbations, as discussed in section V?*

The revised manuscript provides an improved intuitive explanation of the meaning of community "fitness" *F* as an analogue of Tilman's *R** for the case where resources are additive (as considered here), rather than independently required, as in Tilman's classic approach. To illustrate this, Figure 3 now includes an extra panel (Figure 3) showing consistently strengthening resource depletion as equilibrium is approached. In addition, a new section of the Supplementary Material ([Supplementary-material SD1-data], section I) considers the example suggested by the reviewer and explains why the community with the highest *F* will not generally exhibit this kind of "clean resource partitioning", but will include “cross-linkages”.

2) Please at least discuss the sensitivity of the model results to the form of the cost function. For example, how would we expect adding an exponent to norm (σ) in the cost function (2) to change things? This would modulate something like the efficiency of adding additional pathways. Are the conclusions of this work dependent on this assumption that the cost is proportional to the biomass?

The revised manuscript clarifies that the cost function considered here ensures that the cost per pathway ("intrinsic performance measure") is similar for all species. This now reads: "A different cost function would effectively reduce the pool of competitors, excluding certain (prohibitively expensive) species from the start". For example, an exponent other than 1 in Eq. (2) would correspond to a systematic bias favoring generalist or specialist organisms. For strong bias, competition would reduce to the trivial case of a single generalist {1, 1,…,1} or a community of pure specialists, rendering the coalescence problem trivial. A more detailed discussion of this point is now included.

The revised manuscript also explicitly states that the assumption of the biomass being proportional to cost has no effect on conclusions: "a different choice for the biomass of each species would only change transient dynamics, but not the outcome of their competition". The choice made in this work (biomass=cost) is intuitively convenient, since the "intrinsic performance" of a species (measured as equilibrium abundance in a "pure culture" chemostat) then also matches its initial growth rate. A new section was added to the supplementary material to explain this in detail ([Supplementary-material SD1-data], section B).

*3) Throughout, terms could be better defined and explained in the text. In general, the paper is quite heavy in notation, some of which may even be unnecessary. Remind the reader periodically what the variables stand for. For example, each variable could have a name in English and that name could be used together with the symbol in the text. Though it's clear from context, I don't think the paper ever explains in words how σ_i_ is to be interpreted. Nor does g_σ_ seem to be defined. H_i_^(α)^ is not clearly explained; it's an environment, but explain how parametrized. Similarly, the figure and figure legends could be made more self-explanatory. For example, in Figure 1 I had to go back to the main text to determine what epsilon was. This should be explained in words. In the interest of reaching the broadest possible set of readers perhaps consider providing a bit more explanation for technicalities such as what a Lyapunov function is.*

I thank the reviewers for pointing out these presentation issues in such detail. Notations were improved throughout the manuscript (see the PDF of tracked changes); unnecessary notations removed (this includes *P_i_* for pathways, the "star" notation referring to equilibrium quantities, and others). Portions of the argument were converted from formulas into words whenever feasible without ambiguity. *H_i_^(α)^*,*σ*_i_, *g_σ_* are explicitly defined. English names accompany symbols whenever possible. Legends of Figure 2, Figure 3, Figure 5 were expanded to be self-explanatory. References to supplementary file now include section reference. The text was modified so familiarity with the concept of a Lyapunov function is no longer assumed or necessary.